



# Measurement report: size-resolved particle effective density measured by the AAC-SMPS and implications for chemical composition

Yao Song[1], Jing Wei[1], Wenlong Zhao[3], Jinmei Ding[3], Xiangyu Pei[1], Fei Zhang[1], Zhengning Xu[1], Ruifang Shi[1], Ya Wei[1],
Lu Zhang[3], Lingling Jin[3,*], Zhibin Wang[1,2,*]

[1] College of Environmental and Resource Sciences, Zhejiang University, Hangzhou, 310058, China

[2] ZJU-Hangzhou Global Scientific and Technological Innovation Center, Hangzhou, 311200, China

[3] Ecological and Environmental Monitoring Center of Zhejiang Province, Hangzhou 310012, China

*Correspondence to*: Zhibin Wang (wangzhibin@zju.edu.cn) and Lingling Jin (jinlingling@zjemc.org.cn)

**Abstract.** The effective density ($\rho_{\text{eff}}$) is closely associated with the aging process and can serve as a tracer for chemical composition. Recently, studies investigating the effect of particle size on density have been limited. In this study, size-resolved $\rho_{\text{eff}}$ was characterized with a tandem of an aerodynamic aerosol classifier (AAC) and a scanning mobility particle sizer (SMPS) system during one month of observation in Hangzhou. The results indicate that the $\rho_{\text{eff}}$ values of the particles exhibit a unimodal distribution, with average values ranging from 1.47 g/cm³ to 1.63 g/cm³, increasing as the particle diameter increases. The diurnal variation is more pronounced for small particles ($d_{\text{ae}} < 350$ nm), which generally exhibit lower density during the day and higher density at night. The relationship between $\rho_{\text{eff}}$ and particle diameter varies under different pollution conditions due to differences in the chemical composition of the particles. The SHapley additive explanations (SHAPs) revealed good relationships between $\rho_{\text{eff}}$ and the bulk composition of particles with diameters smaller than 350 nm. Since the size-resolved chemical composition of particles is still challenging, a new method to investigate the size-resolved chemical composition was proposed, in which the size-resolved composition can be derived from the $\rho_{\text{eff}}$ and fixed material density of secondary inorganic aerosols (SIAs), organic aerosols (OAs) and black carbon (BC).

## 1 Introduction

Atmospheric aerosols play important roles in human health, air quality, and climate change. The environmental and health impacts of aerosols are largely determined by their chemical and physical properties (Spencer et al., 2007). Density ($\rho$) is one of the most important physical properties of particles. Limited by techniques for measuring $\rho$ for aspherical aerosol particles, $\rho_{\text{eff}}$ has been commonly adopted as an alternative to aerosol density (Peng et al., 2021). It has been largely reported that $\rho_{\text{eff}}$ is related to the particle optical properties (Zhao et al., 2019), aging process (Peng et al., 2016; Leskinen et al., 2023), and chemical composition (Spencer et al., 2007). Therefore, an understanding of size-resolved $\rho_{\text{eff}}$ is important for accurately estimating the effects of particles on climate and health.



Currently, several systems have been developed to measure the size-resolved $\rho_{eff}$. The Micro Orifice Uniform Deposit

Impactor (MOUDI), whose size distribution is measured simultaneously, is a commonly used off-line instrument to measure

size-resolved density (Hu et al., 2012). It collects size-segregated aerosols with different size ranges and subsequently analyses

their mass and chemical components, but the temporal resolution of this offline method is relatively low, and the size range is

limited. The most commonly used method is to measure the mobility diameter and mass simultaneously, after which the $\rho_{eff}$

of the particles can be calculated directly, including the tandem differential mobility analyzer (DMA) and mass analyzer, i.e.,

the aerosol particle mass analyzer (APM) and centrifugal particle mass analyzer (CPMA) (McMurry et al., 2002; Zhou et al.,

2022; Xie et al., 2024). However, the particles are required to be charged before classification for DMA and APM/CPMA,

which introduces multiple charge effects. Particles with higher-order charges can be selected, causing uncertainties in the $\rho_{eff}$

calculation. Additionally, $\rho_{eff}$ can also be derived from the relationship between $d_m$ and (vacuum) aerodynamic diameter ($d_{ae}$),

such as the tandem system of AAC and SMPS (Tavakoli and Olfert, 2014; Song et al., 2022b; Lu et al., 2024) and the tandem

system of ultrafine aerosol time-of-flight mass spectrometry (UF-ATOFMS) and DMA, with the latter also providing size-

resolved chemical composition (Spencer et al., 2007). AAC is a novel method that classifies particles on the basis of their

relaxation time, which can avoid multiple charging effects (Tavakoli and Olfert, 2013). Moreover, the transmission efficiency

of AAC is 2.6-5.1 times greater than that of a combined Krypton85 radioactive neutralizer and DMA (Johnson et al., 2018).

Currently, the AAC-SMPS is commonly used for laboratory-generated particles, but studies on the aerodynamic size-

dependent $\rho_{eff}$ of ambient particles are still limited (Tavakoli and Olfert, 2014; Yao et al., 2020; Kazemimanesh et al., 2022;

Lu et al., 2024).

    The size dependency of particle $\rho_{eff}$ has been reported in previous studies, and the variation in $\rho_{eff}$ is associated with the

chemical composition. Generally, the material densities of inorganic aerosols, such as $(NH_4)_2SO_4$ and $NH_4NO_3$, are greater

than those of OAs. The $\rho_{eff}$ of the particles is greater when the SIA is dominant than when the OA is predominant. The $\rho_{eff}$

increases with the addition of hygroscopic substances, particularly $(NH_4)_2SO_4$ and $NH_4NO_3$ (Yin et al., 2015). The material

density of OAs varies among different sources, but it is greater for secondary organic aerosols (SOAs) than for primary organic

aerosols (POAs) (Zhou et al., 2022). The formation of secondary organic aerosols during active photochemical reactions and

the aging process also leads to an increase in $\rho_{eff}$ (Lin et al., 2018; Xie et al., 2024). The material density of BC is 1.8, but $\rho_{eff}$

can be much smaller because of its fractal morphology. A decrease in $\rho_{eff}$ was found when primary emissions increased, which

led to the accumulation of POA and BC (Lu et al., 2024; Xie et al., 2024).

    However, how the chemical composition influences $\rho_{eff}$ remains unclear because of the complexity of the species

comprising the particles. Currently, machine learning (ML) methods, such as ozone pollution and potential aerosol sources,

are widely used in atmospheric science research (Song et al., 2022a; Cheng et al., 2024; Zhang et al., 2024). ML can identify

the complex and nonlinear relationships between input features and output predictor variables. The SHAP proposed by

Lundberg and Lee (2017) is commonly used to explain the outputs of ML models, which can be used to understand the drivers

of variations in $\rho_{eff}$.



In this study, we present the characteristics of ambient aerosols $\rho_{eff}$ on the basis of one-month online measurements in Hangzhou. The diurnal variation in $\rho_{eff}$ and the influences of the pollution level and air mass transport were discussed. ML and SHAP were used to analyze the relationship between $\rho_{eff}$ and the chemical composition of aerosols. The material densities of OA and BC were determined from the mean $\rho_{eff}$ and bulk chemical composition, and the size-resolved chemical composition was inferred from $\rho_{eff}$ and compared with the bulk chemical composition.

## 2 Experiments

### 2.1 Sampling site and instrumentation

One-month continuous observations were conducted from the 9[th] Sept., 2023, to the 8[th] Oct., 2023, at the Central Air Quality Assurance Monitoring Station for the 19[th] Asian Games in Hangzhou (CASA site). The CASA site (30.25°N, 120.24°E) is located in western Hangzhou city, which is one of the most developed cities in the Yangtze River Delta (YRD) (Fig. S1). The CASA site is near the Qiantang River, which is just 100 m away. There is an elevated road 1.9 km northeast of the site and another 3.2 km southwest. The surrounding area is primarily residential and commercial, with no significant sources of pollution. The local influence at the site is primarily from residential activities and traffic emissions.

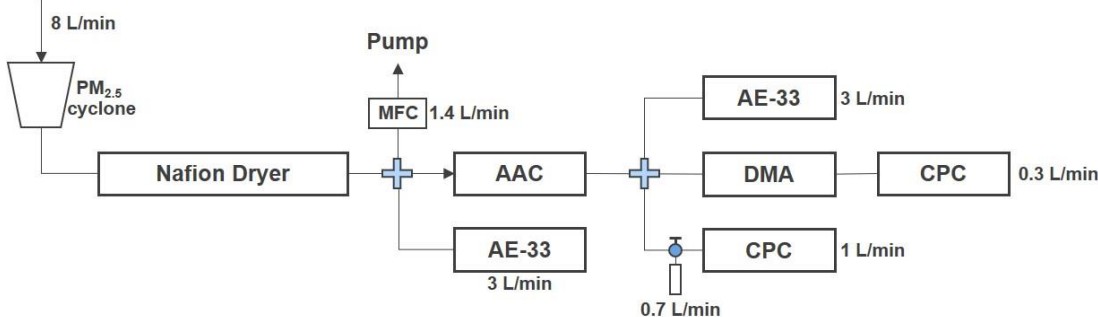

**Figure 1: Instrumental setup used in this study. MFC: Mass flow controller.**

A schematic diagram of the measurement settings is shown in Fig. 1. The particles were sampled with a PM$_{2.5}$ cyclone inlet (8 L/min) and then passed through a Nafion dryer to reduce the relative humidity (RH) to less than 30%. The sampling flow was split into three lines. One line was fed to an AAC-SMPS to determine the effective densities of particles with different diameters. Another line was connected to a seven-wavelength aethalometer (Model AE-33, Magee Scientific, sample flow rate of $Q$ = 3 L/min) to measure the mass concentration of black carbon (BC). The remaining flow was maintained as a bypass flow with an extra pump to ensure an accurate cutoff size. Particles with 12 logarithmically equally distributed aerodynamic sizes ranging from 200-1000 nm were selected by an AAC (Cambustion, Ltd., UK, sheath flow rate $Q_{sh}$ = 9 L/min), and each size was sampled for 5 min. After the AAC classification, the aerosol flow was then split into three parallel lines. One line led to a condensation particle counter (CPC, Model 3772, TSI, Inc., USA) to determine the concentration of the $d_{ae}$-selected particles. The flow rate of the CPC was decreased to 0.3 L/min, with a bypass flow of 0.7 L/min. The second line directed the



flow to an AE-33. The results of the CPC and AE-33 downstream of the AAC will be discussed in other studies. In the remaining line, the particle number size distribution (PNSD) of each $d_{ae}$-selected particle was acquired using an SMPS, and

the modal mobility diameter of the $d_{ae}$-selected particles, denoted as $d_m$, was determined by fitting the PNSD to a log-normal distribution (Eq. S4). However, owing to the low concentrations of larger particles selected by the AAC and the inability to fit the mode $\rho_{eff}$ from $\rho_{eff}$ distributions scanned by the SMPS, only particles with mobility diameters smaller than 600 nm were fitted to lognormal distributions. The bulk chemical composition of nonrefractory aerosols, i.e., the mass concentrations of OA, nitrate, sulfate, ammonium and chloride, was measured via an aerosol chemical speciation monitor (ACSM; Aerodyne

Research, Inc.). Combined with the BC mass concentration, the effective density ($\rho_{eff,cal}$) can be calculated with Eq. 1 (Levy et al., 2013):

$$\frac{1}{\rho_{eff,cal}} = \sum_i \frac{f_i}{\rho_i},$$ (1)

where $\rho_i$ and $f_i$ denote the material density and mass fraction of chemical component i, respectively.

The AAC and SMPS used in this study were calibrated with certified polystyrene latex (PSL) spheres (Thermo Fisher

Scientific Inc., USA) with sizes of 150 nm, 195 nm, 303 nm and 500 nm (Fig. S2). The $d_{ae}$ measured by the AAC was compared with the aerodynamic diameter of the PSL ($d_{ae,PSL}$), which was calculated from the nominal diameter ($d_{m,PSL}$) and material density of the PSL (1.05 g/cm$^3$). The $d_m$ determined by the SMPS was compared to the nominal diameter of the PSL. The deviations between the measured $d_{ae}$ ($d_m$) and $d_{ae,PSL}$ ($d_{m,PSL}$) were 3.19% and 2.48%, respectively. The performance of the AAC-SMPS tandem system was validated by ammonium sulfate (AS) and ammonium nitrate (AN) particles generated with

an atomizer (Model 3079A, TSI, Inc., USA). The effective densities of the size-resolved AS and AN particles are shown in Fig. S2c. The effective densities of AS particles of different sizes are within the range of 1.68–1.82 g/cm$^3$, which is consistent with previous studies (Yao et al., 2020; Lu et al., 2024). The measured average $\rho_{eff}$ of AN particles was 1.80 g/cm$^3$, which is slightly higher than the mean $\rho_{eff}$ of 1.69$\pm$0.17 g/cm$^3$ measured via DMA and a quartz crystal microbalance (QCM) (Sarangi et al., 2016).

**2.2 Machine learning**

The RF models were built with the "RandomForestRegressor" function provided by "scikit-learn" in a python environment (Pedregosa et al., 2011). In the RF models, the independent variables are the size-resolved effective densities of particles with different diameters. The explanatory variables are the mass fractions of the main components of PM$_{2.5}$, i.e., OA, (NH$_4$)$_2$SO$_4$, NH$_4$NO$_3$, NH$_4$Cl and BC. The RF model was trained with a portion of the samples drawn from the whole dataset, while the

remaining samples were used for model validation. To obtain an optimal RF model, the hyperparameters for the RF model were tuned via the function "GridSearchCV" from the "scikit-learn" library (Song et al., 2022a). The coefficient of determination ($R^2$) was used to evaluate the performance of the ML model. The best hyperparameters (n_estimators, min_samples_split, max_features, bootstrap, max-samples) of the RF model and $R^2$ values for particles of different sizes are listed in Table S1. The SHAP approach was implemented via the "shap" python package.





**2.3 Back trajectories**

The regional-scale transport patterns were investigated via back-trajectory calculations with the open-source software Meteoinfo developed by Wang (2019). The model was run 48 h backward every hour from an altitude of 5 m above the site. The NCEP Global Data Assimilation System (GDAS) model with a grid resolution of 1°×1° and a run time of 48 h each day from August to October 2023 was used in the calculations, followed by K-means clustering to investigate sources of air masses, which has been widely used in previous studies (Xu et al., 2023). All the data in this study were in Beijing time.

**3. Results and discussion**

**3.1 Size-resolved effective densities**

The average $\rho_{eff}$ values for particles with aerodynamic diameters of 200, 235, 277, 326, 384, 452 and 531 nm were 1.47±0.09, 1.50±0.09, 1.52±0.08, 1.55±0.08, 1.58±0.08, 1.60±0.08 and 1.63±0.10 g/cm$^3$, respectively (Fig. 2). The distribution of $\rho_{eff}$ was fitted to a unimodal Gaussian distribution, whereas some previous studies reported a bimodal distribution (low and high $\rho_{eff}$ mode) and suggested that the lower peak value was associated with fresh emissions (Qiao et al., 2018; Zhou et al., 2022; Xie et al., 2024). In many studies, the frequency of bimodal distribution is not comparable and largely depends on the distance between the observation points and the emission sources (Zhou et al., 2022; Wu et al., 2023; Xie et al., 2024). The sub-density mode occurrence is greater at sites close to the emission source. The sub-density particles predominated near the street with higher traffic emissions, whereas in the rural background, the main-dense particles dominated (Rissler et al., 2014). A lower frequency of sub-density events indicates that the particles are more likely to be in an internal mixing state. Black carbon, a major product of fossil fuel and biomass combustion, is a tracer of primary emissions (Briggs and Long, 2016). A previous study indicated that when the ratio of the mass concentration of black carbon to PM$_{2.5}$ is less than 20%, particles with diameters between 50 and 350 nm are predominantly in an internal mixing state (Wu et al., 2023). Moreover, the smaller amount of soot particles might be shadowed in the dominant mode of the measurement (Yin et al., 2015). Unfortunately, we did not measure the chemical composition of particles of different sizes, and the bulk chemical composition revealed that the average proportion of BC to PM$_{2.5}$ was only 11% in our study. Therefore, only a unimodal Gaussian fit of $\rho_{eff}$ was identified, indicating that the particles in Hangzhou tend to internally mixed.



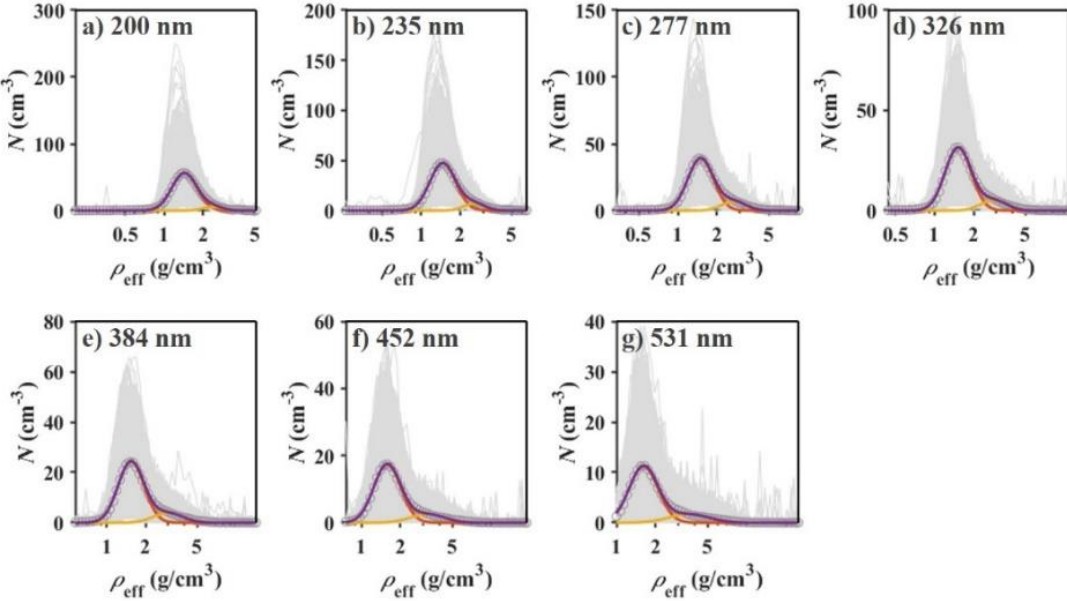

**Figure 2: Averaged particle $\rho_{eff}$ distributions at (a) 200 nm, (b) 235 nm, (c) 277 nm, (d) 326 nm, (e) 384 nm, (f) 452 nm, and (g) 531 nm. The gray lines represent the measured $\rho_{eff}$ distributions. The red and yellow lines are the Gaussian fits of the average $\rho_{eff}$ and the Gaussian fits of doubly charged particles, respectively. The purple line represents the sum of Gaussian fits.**

We also compared the $\rho_{eff}$ values of the particles with those reported in other studies (Fig. 2). Notably, in this study, particles with aerodynamic diameters were selected by the AAC, but generally, the particle mobility diameter was selected in

previous studies. Therefore, the mobility diameters reported in previous studies were first converted to aerodynamic diameters for comparison with the results of our study. The results revealed that the particle $\rho_{eff}$ in most urban regions has a unimodal distribution, with relatively high densities and an average range of 1.2 g/cm$^3$ to 1.8 g/cm$^3$ (Yin et al., 2015; Xie et al., 2017; Lin et al., 2018; Lu et al., 2024). The bimodal distribution of particle $\rho_{eff}$ was found primarily in remote areas and near highways (Rissler et al., 2014; Ma et al., 2017; Ma et al., 2020; Zhou et al., 2022; Xie et al., 2024). Specifically, in the bimodal distribution,

the main-density mode corresponds to a higher $\rho_{eff}$ and increases with increasing particle size, which is similar to that in urban areas. In contrast, the $\rho_{eff}$ of the sub-density mode was generally less than 1 g/cm$^3$ and decreased with increasing particle diameter. The difference between urban and rural sites can be attributed to the mixing state of the particles. Owing to effective secondary organic aerosol formation and emissions from various sources, particles tend to mix internally in urban areas (Ching et al., 2019), leading to a unimodal distribution. Additionally, traffic emissions favor the occurrence of bimodal distributions,

and it has been reported that particles are more externally mixed in street canyons and near highways (Rissler et al., 2014; Riemer et al., 2019). For rural sites, the high frequency of bimodal distribution was determined when the observation site was influenced by local emissions (Zhou et al., 2022).





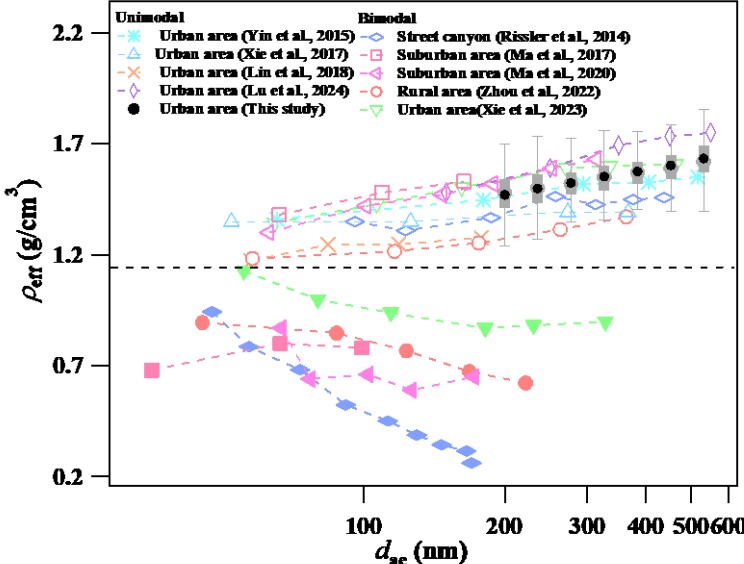

**Figure 3: Size dependency of $\rho_{eff}$ during the observations in this study. The boxes contain the 25th (Q1) and 75th (Q3) boxes. The distance between Q3 and Q1 is the interquartile range (IQR), and the whiskers extend from each quartile to the minimum or maximum. The dots inside the box represent the mean value of $\rho_{eff}$. The size dependency of the effective densities from previous studies is also included for comparison. The black dashed line is drawn to guide the eye.**

### 3.2 Diurnal variations in the effective density

Figure 4 shows the diurnal cycle of $\rho_{eff}$ for particles with diameters of 200 nm and 531 nm, meteorological parameters and mass fractions of different chemical compositions of bulk particles. The diurnal variations in $\rho_{eff}$ of the seven dae-selected particles are shown in Fig. S3. The results indicated that, in comparison with larger particles, smaller particles ($d_{ae}$ <350 nm) exhibit more pronounced diurnal variations in $\rho_{eff}$, typically showing lower values during the day and higher values at night (Fig. 4a). From 7:00 to 9:00, with increasing fresh black carbon (BC) and hydrocarbon-like organic aerosols (HOAs) from vehicle emissions (Fig. 4), the $\rho_{eff}$ of small particles decreases significantly. The volatile and gas–particle partitioning properties of $(NH_4)_2SO_4$ and $NH_4Cl$ are dependent on the ambient T and RH (Sun et al., 2012). Although the atmospheric oxidative capacity increases later (Fig. S4), the photochemical production of $HNO_3$ cannot compensate for evaporative loss under relatively high T conditions (Zhang et al., 2015). The mass fraction of nitrate decreased continuously from 7:00 to 13:00. Moreover, SOA formation increased significantly. As a result, the high material component decreased, whereas OAs with low material density accumulated, leading to a decrease in $\rho_{eff}$. Although the mass fraction of secondary inorganic particles subsequently remains relatively low, ongoing atmospheric aging may cause more small particles to have a spherical morphology (Wang et al., 2017; Wang et al., 2021). This explains why the $\rho_{eff}$ of fine particles gradually increased between 13:00 and 17:00, even though the proportion of high-density components, such as secondary inorganic components, did not significantly increase. After 17:00, the $\rho_{eff}$ values of the small particles did not decrease with increasing BC (Fig. S5f) and POA (Fig. S5g) emissions. Instead, it continued to rise, reaching a peak at 23:00, and then remained relatively stable. At this




time, the increasing RH and decreasing temperature facilitated the redistribution of $HNO_3$ into the particulate phase (Sun et al., 2018; Kuang et al., 2021) and promoted the liquid-phase formation of nitrates on aerosol surfaces (Wang et al., 2020). This led to a dramatic increase in the proportion of SIA, especially for nitrates (Fig. S5d); therefore, the $\rho_{eff}$ of the small particles did not decrease despite the decrease in the concentration of primary emitted particles such as BC and HOA.

In addition, the $\rho_{eff}$ values of larger particles exhibit almost no significant diurnal variation, primarily because larger

particles have longer residence times in the atmosphere and are less influenced by fresh emissions, resulting in a more stable chemical composition (Zhai et al., 2017; Xie et al., 2024). On the other hand, compared with smaller particles, larger particles reduce surface reactions and adsorption capacity due to a lower specific surface area (Okuda, 2013). As a result, the $\rho_{eff}$ of larger particles was less sensitive to changes in temperature and RH.

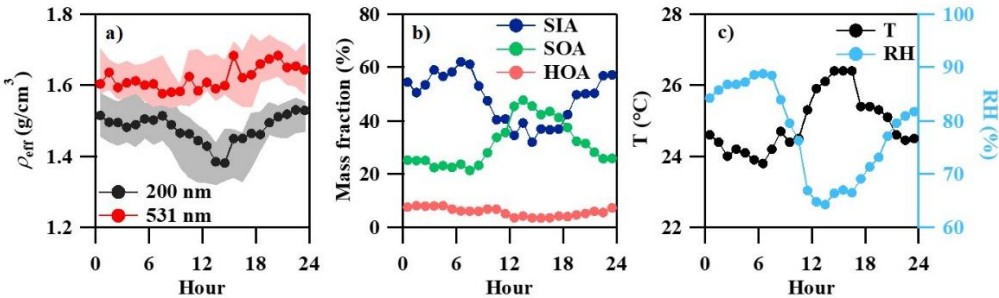

**Figure 4 Diurnal variations in a) size-resolved $\rho_{eff}$, b) mass fractions of the main components and c) metrological parameters during the whole observation period.**

**3.3 Influence of different pollution levels on the particle effective density**

In general, the $\rho_{eff}$ values of particles are influenced mainly by their chemical composition and morphology, which are driven by their atmospheric oxidative capacity, RH, and temperature. Two episodes were identified to further explore the evolution

of the particle $\rho_{eff}$ (Fig. S6). Episode 1 (EP1) and episode 2 (EP2) were run from 18:00 on the 9th to 18:00 on the 12th Sept and from 17:00 on the 23rd to 12:00 on the 30th Sept, respectively, with the rest being the clean period (CP). The $PM_{2.5}$ concentrations of EP1 and EP2 were significantly greater than that of CP (11.4 μg/m³), with average concentrations of 27.9 and 44.7 μg/m³, respectively. EP1 was dominated solely by organic components, which accounted for 64.1±8.0% on average, whereas EP2 was dominated by organic and nitrate components, which accounted for 40.0±13.5% and 29.4±12.9%,

respectively.

The average size-resolved effective densities and bulk chemical compositions under different pollution conditions are shown in Fig. 5. The $\rho_{eff}$ values of all the particles increased with increasing particle size except EP2. Interestingly, the $\rho_{eff}$ of the particles during EP1 (1.47 to 1.61 g/cm³) is smaller than that during CP, which is similar to the observations in the North China Plain (Zhou et al., 2022), indicating that the compositions of particles of the same size are different under different

pollution levels. The proportion of OAs is as high as 64.1%, which is much greater than that during CP (49.3%). Previous studies reported that the effective density of organic materials was between 1.2 and 1.4 g/cm³, which is lower than that of





inorganic materials (Levy et al., 2013). Therefore, the small $\rho_{eff}$ values of all the particles during EP1 could be attributed to the high proportion of OAs. Additionally, air mass cluster analysis indicated that the pollution during EP1 was influenced primarily by regional transport, with a shorter transportation distance than that during CP (Fig. 5d). As a result, the particles had not

fully aged and remained in a relatively fractal state, leading to a lower effective particle density during EP1.

Another interesting phenomenon is that during EP2, the $\rho_{eff}$ of the particles did not increase with increasing particle size. For particles smaller than 400 nm, the average $\rho_{eff}$ (1.51 to 1.55 g/cm$^3$) was greater than that of CP (1.40 to 1.51 g/cm$^3$), whereas for particles larger than 400 nm, the average $\rho_{eff}$ (1.54 to 1.56 g/cm$^3$) was lower than that of CP (1.56 to 1.64 g/cm$^3$). We attempt to explain this phenomenon from two aspects. First, a small particle size results in a large specific surface area,

which promotes the formation of secondary inorganic components. As shown in Figure 4b, the proportion of secondary inorganic compounds during EP2 was as high as 53.2%, which was much higher than the 37.3% reported during CP, with secondary nitrate being the most prominent, whose contribution reached 29.2%. The diurnal variations in size-resolved particles during different periods are shown in Fig. S7. During EP2, from 13:00 to 19:00, the $\rho_{eff}$ of small particles was significantly greater than that during CP. This difference gradually decreased over time, from a peak of 7.3% at 13:00 to 4.7%

at 7:00. This suggests that nitrate formation on small particles was more pronounced during EP2. A previous study revealed that the formation of atmospheric nitrate at high temperatures was due to enhanced heterogeneous processes associated with the aerosol water concentration (Wang et al., 2020). The heterogeneous hydrolysis of $N_2O_5$ was the main mechanism of nitrate formation at night, which was also promoted by high RH and low temperature (Liu et al., 2020). Our observations revealed that the RH and temperature were 81.6% and 26.4°C, respectively, during EP2, which were significantly higher than those of

78.3% and 23.8°C, respectively, during CP. These results are favorable for nitrate formation on small particles during EP2. Additionally, during EP2, the air mass, which passed over areas such as Jiangsu and northern Zhejiang, carried a large number of anthropogenic pollutants, such as $SO_2$. (Fig. 5c). All of these factors resulted in higher precursor concentrations and enhanced the formation of secondary inorganic compounds, particularly on smaller particles. Consequently, the $\rho_{eff}$ of the particles during EP2 remains relatively unchanged with increasing particle size.



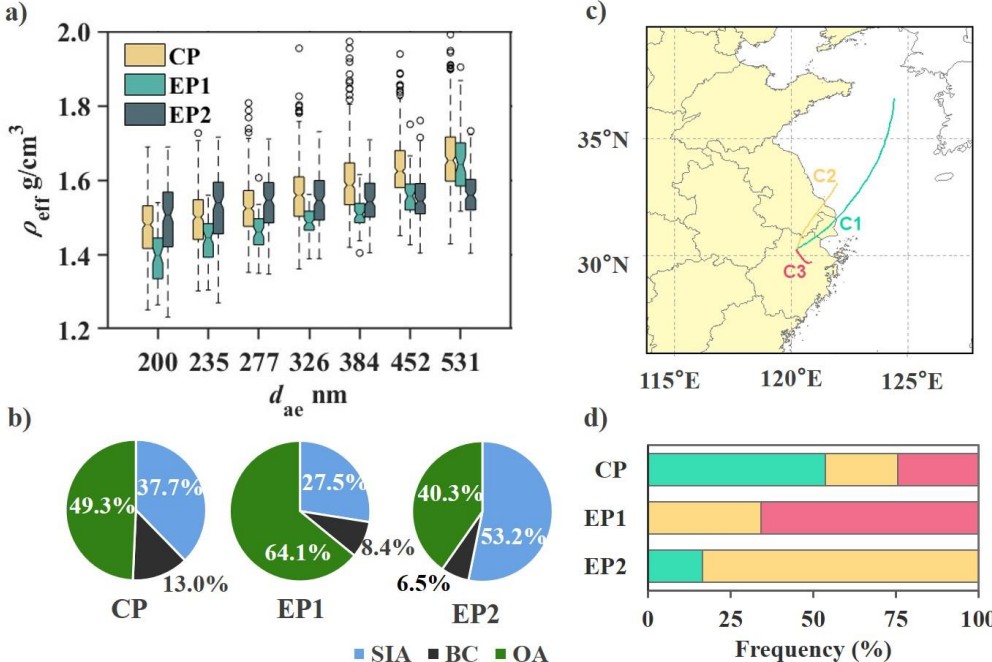


**Figure 5 a) Size-resolved $\rho_{eff}$ of particles with different diameters and b) chemical composition of PM$_{2.5}$ at different pollution levels. (c) Mean 48-h back trajectories of clusters during the measurement period. d) Proportion of different clusters at different pollution levels.**

### 3.4 The influence of chemical composition on the particle effective density

The chemical composition of a particle is the main factor affecting its $\rho_{eff}$. Inorganic components, such as NH$_4$NO$_3$ and (NH$_4$)$_2$SO$_4$, have higher densities than organics and BC. Therefore, variations in aerosol chemical composition can partly explain the observed trend of effective densities. Typically, the size-resolved chemical composition is measured via an aerosol mass spectrometer (Zhang et al., 2005) and single-particle mass spectrometry (Zhai et al., 2017). Due to limitations in measurement techniques, the size-resolved chemical composition of particles is not accurately measured in most studies. Here,

the SHAP method was used to explore the response of different chemical components at every measured particle size on the $\rho_{eff}$ of the particles. Only particles smaller than 350 nm exhibited a strong correlation between feature values and the predicted results, with the correlation coefficient ($R^2$) being greater than 0.55 (Table S1). In contrast, the correlation for larger particles is weaker, possibly because the chemical component variability of larger particles is smaller, resulting in inadequate modeling of these particles. For small particles, organics, NH$_4$NO$_3$, and NH$_4$Cl are the most important chemical components affecting

their $\rho_{eff}$ values (Fig. 6). Specifically, the contribution of the organic proportion was the greatest, with an average contribution of 42.9%. However, this response is negative, indicating that the $\rho_{eff}$ of smaller particles decreases with increasing proportion of organics. NH$_4$NO$_3$ and NH$_4$Cl, which account for 32.2% and 12.7%, respectively, are positively correlated with $\rho_{eff}$,




suggesting that the increase in the proportion of these components mainly leads to a significant increase in the $\rho_{eff}$ of the small particles.

Considering the difficulty of measuring the size-resolved chemical components of particles, this study attempted to calculate the size-resolved composition according to the size-resolved $\rho_{eff}$. According to Eq. 1, the material densities of the ambient OA and BC are needed. For simplicity, the particles were assumed to consist of three components, e.g., OA, SIA, and BC. The value of 1.77 g/cm$^3$ for SIA was adopted in this study. A BC material density in the range of 0.3-2 g/cm$^3$ and an OA material density in the range of 1.2-1.8 g/cm$^3$ were used in the sensitivity test. A step length of 0.02 g/cm$^3$ was adopted for

both the BC density and OA density. The mean $\rho_{eff}$ is the number concentration weighted average density of particles of different sizes. The optimal values of OA and BC were determined by comparing the calculated bulk $\rho_{eff}$ with the average $\rho_{eff}$ for particles with diameters in the range of 200–532 nm. As a result, an OA density of 1.20 g/cm$^3$ and a BC density of 1.77 g/cm$^3$ were determined. The BC density of 1.77 g/cm$^3$ can be attributed to the coating of nitrate since nitrate contributed the most to the mass fraction of PM$_{2.5}$. The $\rho_{eff}$ values of BC were reported to be in the range of 1.62–1.77 g/cm$^3$ for particles with

aerodynamic diameters of 200, 350 and 500 nm in Shanghai, indicating the coating of NH$_4$NO$_3$ and/or (NH$_4$)$_2$SO$_4$ (Wang et al., 2021). The mean $\rho_{eff}$ can be fitted well applying the results of OA and BC, with an R$^2$ of 0.56 (Fig. 7a).

We apply the fitted densities of OA and BC to Eq. 1. Then, the equation is expressed as

$$\frac{1}{\rho_{eff}} = \frac{f_{OA}}{\rho_{OA}} + \frac{f_{BC}}{\rho_{BC}} + \frac{f_{SIA}}{\rho_{SIA}}, \tag{6}$$

where $\rho_{OA}$ is 1.2 g/cm$^3$, $\rho_{BC}$ is 1.77 g/cm$^3$, and $\rho_{SIA}$ is 1.77 g/cm$^3$. $\rho_{eff}$ is the size-resolved $\rho_{eff}$ measured by the AAC-SMPS. A

series of fractions of OA, BC, and SIA were set to calculate $\rho_{eff}$ via Eq. 6, and the fractions of OA, BC, and SIA were adjusted until the difference between the calculated and measured $\rho_{eff}$ values was the minimum value and within an acceptable range (0.5%). Consequently, the scatter plots and time series of the size-resolved proportions of OAs during the observation were determined, as shown in Fig. 7. The slope between the measured and calculated mass fractions of OAs reaches 0.65 for particles with d$_{ae}$<350 nm, indicating that this method can be used to deduce the chemical composition effectively. However, for larger

particles (d$_{ae}$ > 350 nm), the OA and SIA fractions can be reproduced well only during EP2 (Fig. S8), indicating that this simplified method to deduce the chemical composition is not suitable for particles with diameters larger than 350 nm unless the pollution process is dominated by OA and nitrate.





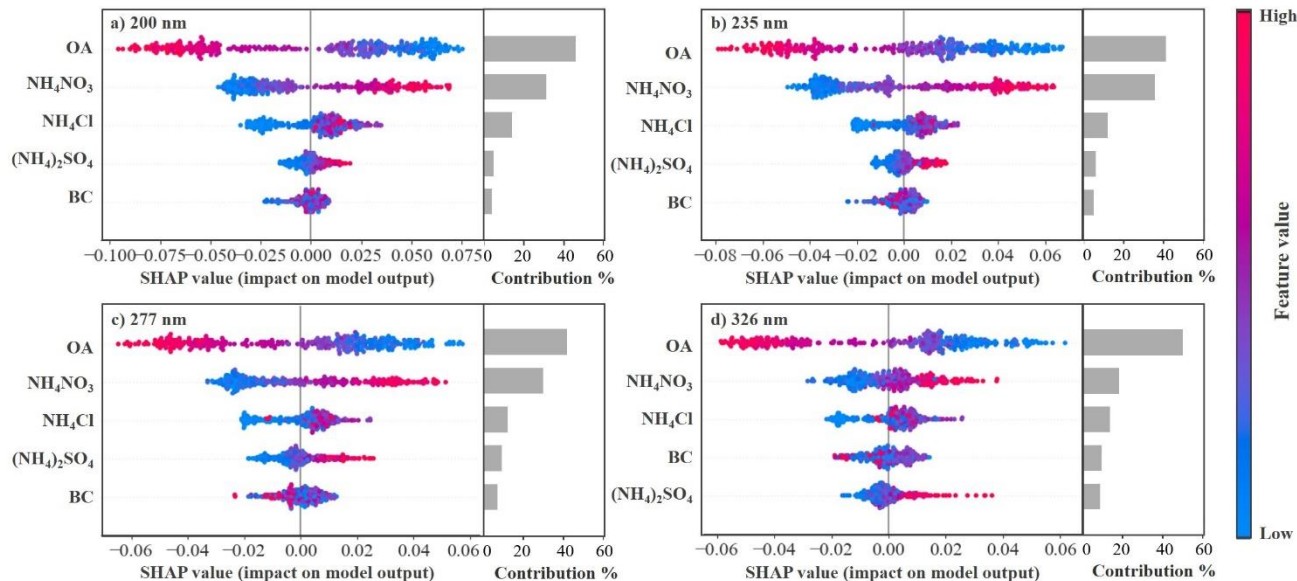

**Figure 6: SHAP values of aerosol composition for particles with diameters of (a) 200 nm, (b) 235 m, (c) 277 nm and (d) 326 nm. The left bar plots attached to each panel describe the contribution of each species.**

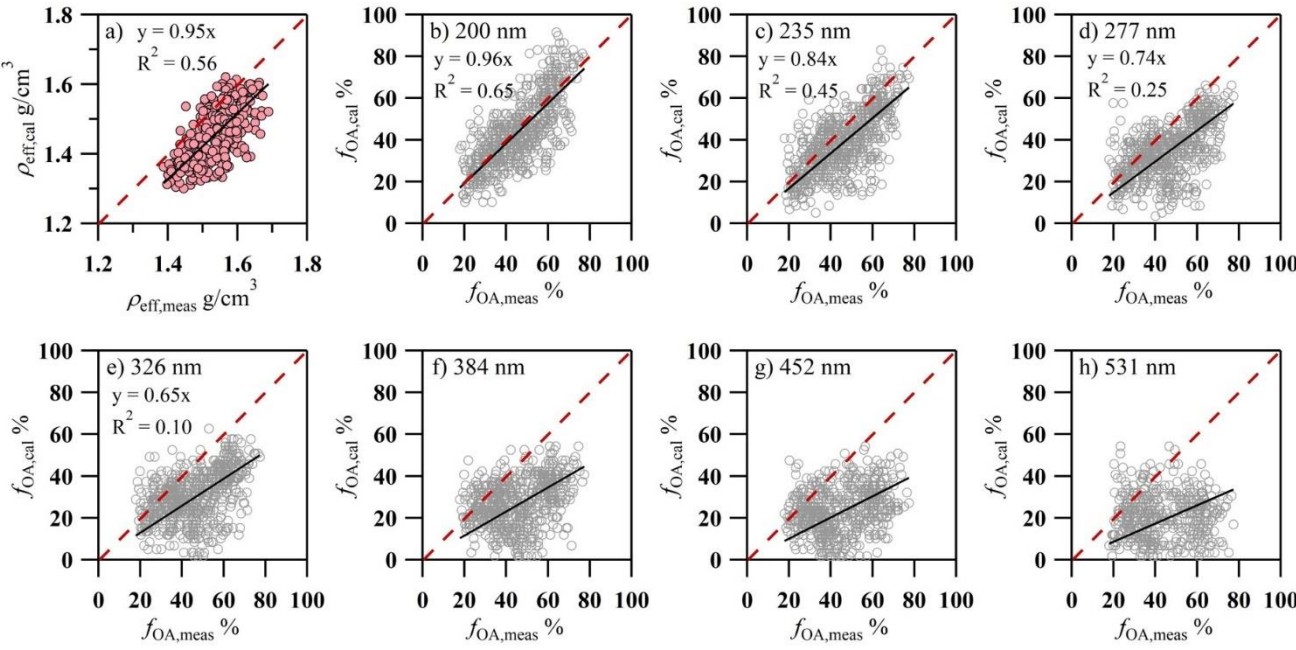

**Figure 7: (a) Comparison of $\rho_{eff}$ calculated from the ACSM aerosol composition ($\rho_{eff,cal}$) and the mean $\rho_{eff}$ measured by the AAC-SMPS ($\rho_{eff,meas}$). Comparison of the calculated mass fraction of OAs from the measured $\rho_{eff}$ and the measured mass fraction of OAs with an ACSM for particles with diameters of (b) 200 nm, (c) 235 nm, (d) 277 nm, (e)326 nm, (f) 384 nm, (g) 452 nm, and (h) 531 nm. The red lines denote the 1:1 line.**



## 4. Conclusion

The effective density is one of the most important physical properties and is associated with the aging process and chemical composition of aerosols. In this study, the AAC-SMPS was used to characterize the size-resolved $\rho_{eff}$ in autumn in Hangzhou. The measured $\rho_{eff}$ distributions were fitted to unimodal log-normal distributions, and the size dependency of $\rho_{eff}$ was determined.

The $\rho_{eff}$ varied from 1.47±0.09 to 1.63±0.10 g/cm$^3$ as the diameter increased from 200 nm to 531 nm. The $\rho_{eff}$ of small particles ($d_{ae}$<350 nm) shows more pronounced diurnal variation due to changes in chemical composition and particle morphology, with lower values during the day and higher values at night, whereas the $\rho_{eff}$ of larger particles shows no significant diurnal cycle. Moreover, $\rho_{eff}$ is associated with pollution levels and air mass transport. During the clean period, $\rho_{eff}$ is relatively high and clearly depends on size. However, during pollution episodes, the size dependence of $\rho_{eff}$ is significantly

weakened. Particularly when the pollutant components are dominated by organics and secondary nitrates, the $\rho_{eff}$ of the particles does not increase with increasing particle size.

ML and SHAP were used to analyze the relationship between $\rho_{eff}$ and the chemical composition of aerosols. OA, NH$_4$NO$_3$ and NH$_4$Cl have the strongest effects on $\rho_{eff}$. $\rho_{eff}$ has a negative correlation with OA and positive correlations with NH$_4$NO$_3$ and NH$_4$Cl. Assuming that the particles consisted of OAs, SIAs and BCs, the material density of OAs of 1.2 g/cm$^3$ and that of

BCs of 1.7 g/cm$^3$ were derived from the mean $\rho_{eff}$ and bulk chemical composition. Using the derived material, the size-resolved chemical composition was inferred from the size-resolved $\rho_{eff}$. The composition can be reproduced well for small particles, but this simplified method is not suitable for particles with diameters larger than 350 nm unless the pollution process is dominated by OA and nitrate.


**Data availability.** The data are available from the data repository of Zenodo at http://doi.org/10.5281/zenodo.13981448 (Song et al., 2024).

**Author contributions.** ZW determined the main goal of this study and designed this research. YS, JW and XP conducted the

field measurements with support from RS, YW, WZ and JD. YS and JW performed the data analysis. JW, XP, FZ, ZX, LZ and LJ contributed to the data analysis and interpretation. The written article was prepared by YS, JW and ZW with input from all other coauthors.

**Competing interests.** At least one of the (co-)authors is a member of the editorial board of ACP.

**Financial support.** This work was supported by the National Key Research and Development Program of China (Grant No.

2022YFC3703505); the National Natural Science Foundation of China (Grant No. 42005086, 42305098); the China Postdoctoral Science Foundation (Grant No. 2023M733028); the Postdoctoral Fellowship Program of CPSF (Grant No. GZC20232276); "Pioneer" and "Leading Goose" R&D Program of Zhejiang (Grant No. 2022C03065).





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
