# Peer review of "Measurement report: size-resolved particle effective density measured by the AAC-SMPS and implications for chemical composition"

_EGUsphere, 2024_

## Author Comment (AC1)

**Response to Anonymous Referee #1**

We thank the reviewer for the constructive suggestions and comments concerning our manuscript entitled "Measurement report: size-resolved particle effective density measured by the AAC-SMPS and implications for chemical composition" (ID: egusphere-2024-3298). Those comments are valuable and very helpful for improving our paper, as well as the important guiding significance to our studies. Below, we provide a point-by-point response to individual comment (Reviewer comments in italics, responses in plain font; page numbers refer to the ACPD version; Tables used in the response are labeled as Table R1, Table R2,…, figures used in the response are labeled as Fig. R1, Fig. R2,…)

**[Comments1]** *The study investigated size-resolved particle effective density and its relationship with chemical composition, which has implications for air quality, climate, and health. The use of AAC-SMPS in tandem with machine learning (ML) techniques, such as SHAP analysis, demonstrates a commendable level of innovation. However, I have two concerns: (a) first, I am somewhat worried about the accuracy of the measurements from the entire system. (b) Second, one of the highlights of the paper is using size-resolved effective density to infer particle composition, but the paper does not provide composition information for different sizes for comparison. I recommend a major revision of the paper.*

**Responses and Revisions:**

(a) Thank you for the advice. This system has been validated with PSL, ammonium sulfate and ammonium nitrate aerosols. The validation with PSL aerosols was discussed in Comments2 and the uncertainty analysis was shown in Comments3. The effective density of PSL particles determined with this system was 1.045 g/cm3, which was consistent with the material density of PSL (1.05 g/cm$^3$). Besides, the uncertainty of effective density measured in this study was within the range of 3.00%-3.05%. In conclusion, measurement of effective density with this AAC-SMPS system is reliable.

(b) The size-resolved chemical composition is usually measured via an aerosol mass spectrometer and single-particle mass spectrometry. Due to limitations in measurement techniques, the size-resolved chemical composition of particles hasn't been characterized in this study. However, Previous studies on size-resolved chemical composition of particles have shown that the proportion of inorganic salts increases with particle size (Zhang et al., 2005; Kim et al., 2020; Zhao et al., 2020). The effective density of inorganic salts is approximately 1.77 g/cm³, causing the effective density of large particles to approach that of inorganic salts. Consequently, the effective density of large particles does not vary significantly with changes in chemical composition. Meanwhile, the analyses of effective density variations and SHAP (SHapley Additive exPlanations) results have demonstrated a significant correlation between the effective density of small particles ($d_{ae}$<350 nm) and their overall chemical composition. Therefore, it is feasible to infer the chemical composition of small particles ($d_{ae}$<350 nm) based on their effective density. We must acknowledge that this calculation would be more rigorous if size-resolved chemical composition data were available. Here, we propose this method, which remains applicable if size-resolved chemical composition measurements become

available in future studies. Besides, we also conducted a sensitivity analysis, which revealed that the fitting results are more sensitive to the density of BC than to that of OA (Fig R1). For 200 nm particles, when the BC density is 1.77 g/cm³, the fitting results exhibit a higher tolerance for variations in the OA density. However, as particle size increases, this tolerance decreases. Therefore, when inferring chemical composition from effective density, it is more important to accurately constrain the densities of OA and BC for larger particles. The sensitivity analysis has been added in manuscript (Lines 291-297).

[Figure]

**Figure R1: the fitting coefficients of the calculated mass fraction of OAs from the measured $\rho_{eff}$ and the measured mass fraction of OAs with an ACSM for particles with diameters of (a) 200 nm, (b) 235 nm, (c) 277 nm. The OA mass fraction was calculated with $\rho_{OA}$ of 1.0-1.6 g/cm³ and $\rho_{BC}$ of 0.5-2.5 g/cm³.**

*Comments:*

**[Comments2]** *(a)Why does the effective density of ammonium sulfate increase with particle size, while ammonium nitrate remains relatively stable (Fig. S2)? (b) I'm curious about the measurement results of this system for the effective density of PSL (polystyrene latex) spheres with different particle sizes. Since PSL spheres are essentially regular spherical shapes, their effective density should theoretically equal the material density (1.05 g/cm³). If the effective density of PSL spheres with different sizes cannot remain stable, I suspect that the observed relationship between effective density and particle size may not reflect the true situation but rather a bias in the observation system.*

**Responses and Revisions:**

Thank you for the advice.

(a) The tendency of size-resolved density of ammonium sulfate (AS) measured in this study is consistent with the results in previous studies, which is shown in Fig. R2. In general, the effective density of AS particles increases with particle size, which can be attributed to the change of morphology. Previous studies have reported various morphologies of ammonium sulfate particles, including dome-like, ball-like, redundant-rectangular, and clustered particles (Ueda, 2021). The AS particles tend to be spherical with the increasing size, leading to the larger effective density. The size-resolved effective density results for ammonium nitrate (AN) are lacked. However, the commonly used bulk density for ammonium nitrate is 1.7 g/cm³ (Neuman et al., 2003; Sarangi et al., 2016), which is

consistent with our findings. The weak size dependence of AN effective density may be attributed to the relatively stable morphology of particles across different sizes.

[Figure]

**Figure R2: The size-resolved effective density of ammonium sulfate measured in previous studies and this study.**

(b)  We also used PSL particles with nominal diameters of 100, 150, 200, 300, and 500 nm to validate the accuracy of this system. The result has been added in Lines 108-110:

"The performance of the AAC-SMPS tandem system was first validated with PSL particles and the average effective density of 1.045 g/cm³ was determined, which is consistent with the material density of PSL (1.05 g/cm³) (Fig. S2)."

The results of PSL effective density measured with the AAC-SMPS was shown in Supplement Fig S2:

[Figure]

**Figure S2: Size-resolved $\rho_{eff}$ values of PSL particles measured by AAC-SMPS**

The measured results of the effective density of PSL particles indicate that the AAC-SMPS provides reliable results for aerosol effective density, making it suitable for subsequent analysis.

**[Comments3]** *Line 85 (a) Please specify the resolution ($R_s$) of AAC, the sheath of DMA, and the scanning settings of SMPS (such as diameter range, time for one scan, how many SMPS scans in 5 min). The Rs and DMA sheath could both influence the uncertainty of the results. (b) Besides, After the authors provide these parameters, it is recommended that they perform an uncertainty analysis.*

**Responses and Revisions:**

Thank you for the advice. When selecting particles with an AAC, the sheath flow rate was maintained unchanged to keep the relaxation time resolution $R_t = Q_{sh}/Q_a = 2.5$ as a constant. The aerodynamic size resolution $R_s$ varies with the nominated $d_{ae}$ and it can be calculated as follows,

$$R_s = R_t / \left( \frac{Cc(d_{ae})}{\frac{dCc(d_{ae})}{dd_{ae}} + 2Cc(d_{ae})} \right), \tag{R1}$$

in which $Cc(d_{ae})$ is the Cunningham slip correction factor, $d_{ae}$ is the nominated aerodynamic diameter.

a) The settings of SMPS has been added in Line 90:

"the particle number size distribution (PNSD) of each $d_{ae}$-selected particle was acquired using an SMPS, which was consisted of a soft X-ray neutralizer (Model 3088, TSI Inc., USA), a DMA (Model 3081, TSI Inc., USA., sheath flow rate $Q_{sh} = 3$ Lpm) and a CPC (Model 3756, TSI Inc., USA., sample flow rate $Q_a = 0.3$ Lpm). The mobility diameter range was 13.8-749.9 nm with a total scan time of 4 minutes."

(b) The uncertainty analysis has been added in the Supplement Sect. S1:

"The aerodynamic dia`meter can be calculated from the particle relaxation time according to Eq. S2. Applying the propagation of uncertainty, the uncertainty of $d_{ae}$ can be derived as follows,

$$\left( \frac{\varepsilon_{d_{ae}}}{d_{ae}} \right)^2 = \frac{1}{4} \left( \frac{\varepsilon_\tau}{\tau} \right)^2 + \frac{1}{4} \left( \frac{\varepsilon_\mu}{\mu} \right)^2 + \frac{1}{4} \left( \frac{\varepsilon_{Cc}}{Cc} \right)^2, \tag{S5}$$

where $\varepsilon_\mu/\mu = 1.2\%$, $\varepsilon_{Cc}/Cc$ is the same for all particle sizes and equals 2.1%, and $\varepsilon_\tau/\tau$ is associated with the sheath flow rate $Q_{sh}$, rotating rate $\omega$ and dimensional parameters (length $L$ and mean radius of inner and outer cylinders $\bar{r}$) of AAC

$$\left( \frac{\varepsilon_\tau}{\tau} \right)^2 = \left( \frac{\varepsilon_{Q_{sh}}}{Q_{sh}} \right)^2 + 4 \left( \frac{\varepsilon_\omega}{\omega} \right)^2 + 4 \left( \frac{\varepsilon_{\bar{r}}}{\bar{r}} \right)^2 + \left( \frac{\varepsilon_L}{L} \right)^2, \tag{S6}$$

where $\varepsilon_{Q_{sh}} = 0.1$ Lpm, $\varepsilon_\omega = 5$ rpm, $\varepsilon_L = 2$ mm, and $\varepsilon_{\bar{r}} = 5$ μm.

According to Eq. S1, the uncertainty for particle mass is

$$\left( \frac{\varepsilon_m}{m} \right)^2 = \left( \frac{\varepsilon_\tau}{\tau} \right)^2 + \left( \frac{\varepsilon_B}{B} \right)^2, \tag{S7}$$

and $\frac{\varepsilon_B}{B}$ can be written as follows according to Eq. S2,

$$\left( \frac{\varepsilon_B}{B} \right)^2 = \left( \frac{\varepsilon_{Cc}}{Cc} \right)^2 + \left( \frac{\varepsilon_\mu}{\mu} \right)^2 + \left( \frac{\varepsilon_{d_m}}{d_m} \right)^2, \tag{S8}$$

where $\frac{\varepsilon_{d_m}}{d_m} = 3\%$.

As a result, the uncertainty in effective density is

$$\left(\frac{\varepsilon_{\rho_{eff}}}{\rho_{eff}}\right)^2 = 9\left(\frac{\varepsilon d_m}{d_m}\right)^2 + \left(\frac{\varepsilon_m}{m}\right)^2. \tag{S9}$$

As the sheath flow rate used in this study is a constant, there is not much difference among the uncertainty for effective densities of selected particles with different sizes (Table S1).

**Table S1: the uncertainty for effective densities of selected particles with different sizes measured by the AAC-SMPS.**

| $d_{ae}$(nm) | $\tau(\%)$ | $\rho(\%)$ |
|---|---|---|
| 200 | 1.50 | 3.00 |
| 235 | 1.51 | 3.00 |
| 277 | 1.52 | 3.01 |
| 326 | 1.52 | 3.01 |
| 384 | 1.54 | 3.02 |
| 452 | 1.57 | 3.03 |
| 531 | 1.59 | 3.05 |

"

**[Comments4]** *Line 98 The effective density of BC (could be smaller than 0.5 g/cm³ for fresh BC) is totally different from BC's material density (~1.8 g/cm³), is it reasonable to use the material density of black carbon to calculate the overall effective density?*

**Responses and Revisions:**

We apologize for the misunderstanding. the density used in this equation is volume equivalent density with void instead of the material density. Previous studies have reported effective density of BC varies within a large range of 0.1 – 1.8 g/cm³ (Zhou et al., 2022). Here in Line 98, we didn't define the specific values of effective density of BC and OA. To determine the correct BC density in this equation, a BC material density in the range of 0.3-2 g/cm³ was used in the sensitivity test in section 3.4. This sentence has been revised as:

"$\rho_i$ and $f_i$ denote the effective density and mass fraction of chemical component $i$, respectively. The values of effective density of BC and OA are discussed in Section 3.4" (Line 102-103).

**[Comments5]** *Line 128-143 Another possibility for the unimodal distribution is that the minimum particle size setting for this observation was $D_{ae}$ = 200 nm, and the aerodynamic diameter of most fresh black carbon particles is smaller than this, so the bimodal distribution could not be detected.*

**Responses and Revisions:**

Thank you for the advice. This possibility has been added in the analysis, and Line 136-140 has been revised as:

"Black carbon, a major product of fossil fuel and biomass combustion, is a tracer of primary emissions (Briggs and Long, 2016). On the one hand, the bimodal distribution was not detected may because of the absence of fresh BC. The diameter of fresh BC mainly ranged from 50-120 nm (Bond et al., 2013), which was much smaller than the selected particles in our study. On the other hand, a previous study indicated that when the ratio of the mass concentration of black carbon to $PM_{2.5}$ is less than 20%, particles with diameters between 50 and 350 nm are

predominantly in an internal mixing state (Wu et al., 2023). Moreover, the smaller amount of soot particles might be shadowed in the dominant mode of the measurement (Yin et al., 2015)."

**[Comments6]** *Line 165 Please improve the clarity of Figure 3, especially as the figure legend is a bit blurry.*

**Responses and Revisions:**

Thank you for the advice. This figure has been replotted.

**[Comments7]** *Line 190 I have two questions regarding the measurement and discussion of large particles ($D_{ae}$ = 531 nm):*
*(a) It was mentioned earlier (line 92) that particles with a mobility diameter exceeding 600 nm were not used in Gaussian fitting. How significant is the impact of this on the effective density determination at 531 nm?*
*(b) The number concentration of large particles is relatively small. After AAC selection, can the size distribution measured by SMPS still be fitted with a Gaussian function? I would like to see the size distribution scans by SMPS at different $D_{ae}$ values."*

**Responses and Revisions:**

(a) The averaged size distribution of particles with $d_{ae}$ of 531 nm during the observation is shown as Fig. R3g. Apart from the relatively low concentration, the inability to fit particle sizes above 600 nm is due to the fitted Gaussian distribution exceeding the measured size range. Although the main Gaussian distribution range fitted for 531 nm particles also partially exceeds the measurement range, we believe this does not lead to errors in the modal value. In log-normal distribution, the 95.4% data points fall within twice standard deviations of the mean ($2\sigma$). The maximum value of measured particle size is 495.8 nm, which is comparable with the $2\sigma$ (526 nm) of the main peak. Therefore, even if the main peak is not fully captured, we still consider the fitted particle size to be accurate.

[Figure]

**Figure R3: The averaged size distribution of particles at (a) 200 nm, (b) 235 nm, (c) 277 nm, (d) 326 nm, (e) 384 nm, (f) 452 nm, and (g) 531 nm. The gray lines represent the measured size distribution. The red and yellow lines are the Gaussian fits of singly charged particles and the Gaussian fits of doubly charged particles, respectively. The purple line represents the sum of Gaussian fits.**

(b) The size distribution can be fitted to bimodal Gaussian functions with a sub-model as the particles with double charges. Due to the large number of measured PNSD during the observation period (6370 distributions), Fig. R3 only presents the Gaussian fits for all size distributions and the average size distribution. We conducted a statistical analysis of all the Gaussian fitting results. The $R^2$ and the two model values of bimodal Gaussian fitting are shown in Fig. R4. The median values of fitting coefficient $R^2$ for particles with $d_{ae}$ of 200-531 nm are 0.99, 0.98, 0.98, 0.97, 0.96, 0.95 and 0.92, respectively. The results showed that a log-normal distribution provides a good fit for the number size distributions of particles across different sizes. Even for the 531 nm particles, the median fit coefficient was 0.92.

[Figure]

**Figure R4: The $R^2$ and the two model values of bimodal Gaussian fitting.**

**Reference**

Bond, T. C., Doherty, S. J., Fahey, D. W., Forster, P. M., Berntsen, T., DeAngelo, B. J., Flanner, M. G., Ghan, S., Kärcher, B., Koch, D., Kinne, S., Kondo, Y., Quinn, P. K., Sarofim, M. C., Schultz, M. G., Schulz, M., Venkataraman, C., Zhang, H., Zhang, S., Bellouin, N., Guttikunda, S. K., Hopke, P. K., Jacobson, M. Z., Kaiser, J. W., Klimont, Z., Lohmann, U., Schwarz, J. P., Shindell, D., Storelvmo, T., Warren, S. G., and Zender, C. S.: Bounding the role of black carbon in the climate system: A scientific assessment, J. Geophys. Res.: Atmos., 118, 5380-5552, https://doi.org/10.1002/jgrd.50171, 2013.
Kim, N., Yum, S. S., Park, M., Park, J. S., Shin, H. J., and Ahn, J. Y.: Hygroscopicity of urban aerosols and its link to size-resolved chemical composition during spring and summer in Seoul, Korea, Atmos. Chem. Phys., 20, 11245-11262, https://doi.org/10.5194/acp-20-11245-2020, 2020.

Neuman, J. A., Nowak, J. B., Brock, C. A., Trainer, M., Fehsenfeld, F. C., Holloway, J. S., Hübler, G., Hudson, P. K., Murphy, D. M., Nicks Jr., D. K., Orsini, D., Parrish, D. D., Ryerson, T. B., Sueper, D. T., Sullivan, A., and Weber, R.: Variability in ammonium nitrate formation and nitric acid depletion with altitude and location over California, J. Geophys. Res.: Atmos., 108, https://doi.org/10.1029/2003JD003616, 2003.

Sarangi, B., Aggarwal, S. G., Sinha, D., and Gupta, P. K.: Aerosol effective density measurement using scanning mobility particle sizer and quartz crystal microbalance with the estimation of involved uncertainty, Atmos. Meas. Tech., 9, 859-875, https://doi.org/10.5194/amt-9-859-2016, 2016.

Ueda, S.: Morphological change of solid ammonium sulfate particles below the deliquescence relative humidity: Experimental reproduction of atmospheric sulfate particle shapes, Aerosol Sci. Technol., 55, 423-437, https://doi.org/10.1080/02786826.2020.1864277, 2021.

Zhang, Q., Canagaratna, M. R., Jayne, J. T., Worsnop, D. R., and Jimenez, J. L.: Time- and size-resolved chemical composition of submicron particles in Pittsburgh: Implications for aerosol sources and processes, J. Geophys. Res.: Atmos., 110, https://doi.org/10.1029/2004jd004649, 2005.

Zhao, P., Du, X., Su, J., Ding, J., and Dong, Q.: Aerosol hygroscopicity based on size-resolved chemical compositions in Beijing, Sci. Total Environ., 716, 137074, https://doi.org/10.1016/j.scitotenv.2020.137074, 2020.

Zhou, Y., Ma, N., Wang, Q., Wang, Z., Chen, C., Tao, J., Hong, J., Peng, L., He, Y., Xie, L., Zhu, S., Zhang, Y., Li, G., Xu, W., Cheng, P., Kuhn, U., Zhou, G., Fu, P., Zhang, Q., Su, H., and Cheng, Y.: Bimodal distribution of size-resolved particle effective density: results from a short campaign in a rural environment over the North China Plain, Atmos. Chem. Phys., 22, 2029-2047, https://doi.org/10.5194/acp-22-2029-2022, 2022.

---

## Author Comment (AC2)

**Response to Anonymous Referee #2**

We thank the reviewer for the constructive suggestions and comments concerning our manuscript entitled "Measurement report: size-resolved particle effective density measured by the AAC-SMPS and implications for chemical composition" (ID: egusphere-2024-3298). Those comments are valuable and very helpful for improving our paper, as well as the important guiding significance to our studies. Below, we provide a point-by-point response to individual comment (Reviewer comments in italics, responses in plain font; page numbers refer to the ACPD version)

The current manuscript presents the characteristics of the ambient aerosol density of one-month online measurements in Hangzhou. The ms is generally written well. I have mainly questions for clarification.

**[Comments1]** *Do the SHAPs work for particles larger than 350 nm? If not, it deserves to be mentioned in the abstract.*

**Responses and Revisions:**

Thank you for the advice. According to the SHAP results, the correlation between the effective density and the chemical composition of particles with diameter larger than 350 nm was weak. The Line 17 in abstract has been revised as

"The SHapley additive explanations (SHAPs) revealed good relationships between $\rho_{eff}$ and the bulk composition of particles with diameters smaller than 350 nm, while the relationship of larger particles was weak."

**[Comments2]** *Lines 33–34: Please specify the size range.*

**Responses and Revisions:**

Thank you for the advice. This sentence has been revised as

"It collects size-segregated aerosols with different size ranges and subsequently analyses their mass and chemical components, but the temporal resolution of this offline method is relatively low, and the size range is limited (0.056 – 18 μm, depending on size stages of used MOUDI)" (Line 34)

**[Comments3]** *The current system uses a dryer before the instruments. When RH is high enough, ambient particles may exist as aqueous droplets. Is there any effect of the particle phase state on density measurements?*

**Responses and Revisions:**

Previous studies have indicated that aerosol particles can absorb and release water when they undergo relative humidity (RH) cycles which can govern the liquid water content, composition, size and phase state (liquid, semisolid, or solid) of aerosol particles (Tan et al., 2024). Liquid water has a much lower density (approximately 1 g/cm³) compared to solid particles like salts or organic compounds (which can range from 1.5 to 2.5 g/cm³ depending on the material). The uptake of water can lead to a lower overall density of particles.

In order to avoid the uncertainty of changing RH, we think the dryer before the instrument is necessary. When the particles were introduced into the instrument, it is difficult to keep the RH inside the instrument same as the ambient RH. Different RH can affect the water content of aerosols. Drying the particles standardizes the measurement conditions, making it easier to compare aerosol properties across different sites, times, and studies. Without drying, the measurements could be influenced by varying humidity levels, introducing inconsistencies.

[**Comments4**] *Please define the sub-density mode.*

**Responses and Revisions:**

Thank you for the advice. The definition of sub-density has been elaborated in Line 136-140:

"The distribution of $\rho_{eff}$ was fitted to a unimodal Gaussian distribution, whereas some previous studies reported a bimodal distribution, i.e., a mode with higher peak value and larger effective density denoted as main-density mode and another mode with lower peak value and lower effective density denoted as sub-density mode. The sub-density mode was associated with fresh emissions (Qiao et al., 2018; Zhou et al., 2022; Xie et al., 2024)"

[**Comments5**] *The relationship between density distribution and particle mixing state is unclear. Please elaborate on what the bimodal distribution means.*

**Responses and Revisions:**

Thank you for the advice. It has been added in Line 131:

"The unimodal distribution denoting an internally mixed aerosol composition and the bimodal distribution with a second, below unity density peak indicating externally mixed BC (Qiao et al., 2018; Ma et al., 2020; Zhou et al., 2022; Xie et al., 2024)."

[**Comments6**] *The authors conclude no significant diurnal variation in the density values. However, Figure 4a noted a gradual increase in the density for 531 nm particles, which deserves more discussion and comparisons with other studies.*

**Responses and Revisions:**

Thank you for the advice. The diurnal variation for larger particles has been added in Line 189.

"In addition, compared to small particles, the diurnal variation in effective density of large particles is less pronounced. The decreasing trend during 7:00 to 13:00 becomes less obvious (Fig. S4), primarily because larger particles have longer residence times in the atmosphere and are less influenced by fresh emissions, resulting in a more stable chemical composition (Zhai et al., 2017; Xie et al., 2024). Overall, the effective density of large particles shows a slight increase throughout the day. It would be related to the increasing of RH at night. which facilitates the formation of SIA which has been discussed before. However, compared with smaller particles, larger particles reduce surface reactions and adsorption capacity due to a lower specific surface area (Okuda, 2013). As a result, the $\rho_{eff}$ of larger particles was less sensitive to changes in temperature and RH."

**[Comments7]** *In this study, only small particles showed a strong correlation between feature values and the prediction. (a) Please specify the feature values. (b) Does the weak correlation for big particles mean that the chemical composition is invariable compared to that of small particles?*

**Responses and Revisions:**

(a) Feature values estimate the significance of each feature within a model. This sentence has been revise as:

"The cross-validation was used to evaluate the model performance, and the results suggested that RF model was performing well for particles smaller than 350 nm, with the overall $R^2$ score being greater than 0.55 (Table S1)." (Line 261)

(b) Yes, the weak correlation suggested the relatively stable chemical composition. The inorganic salts dominate in these particles. Previous studies on size-resolved chemical composition of particles have shown that the proportion of inorganic salts increases with particle size (Zhang et al., 2005; Kim et al., 2020; Zhao et al., 2020). The effective density of inorganic salts is approximately 1.77 g/cm³, causing the effective density of large particles to approach that of inorganic salts.

**Minor comments:**

**[Comments8]** *Line 58: Please rewrite "machine learning (ML) methods, such as ozone pollution and potential aerosol sources." Ozone pollution and potential aerosol sources are not ML methods.*

**Responses and Revisions:**

Thank you for the advice. It has been revised as (Lines 58-59):
"Currently, machine learning (ML) methods, such as simulation of ozone concentration and reproduction of aerosol number concentration, are widely used in atmospheric science research"

**[Comments9]** *Line 111: Please define "RF."*

**Responses and Revisions:**

Thank you for the advice. It has been added in introduction (Line 60):
"Random forest (RF) is a commonly-used machine learning algorithm. Compared to other ML models, such as deep learning models, it maintains a commendable balance between predictive performance and interpretability."

**[Comments10]** *Line 148: Fig.2 -> Fig.3.*

**Responses and Revisions:**

Thank you for the advice. It has been revised.

Reference

Tan, F., Zhang, H., Xia, K., Jing, B., Li, X., Tong, S., and Ge, M.: Hygroscopic behavior and aerosol chemistry of atmospheric particles containing organic acids and inorganic salts, npj Clim Atmos Sci, 7, 203, https://doi.org/10.1038/s41612-024-00752-9, 2024.